# Factors Influencing Construction Waste Generation in Building Construction: Thailand's Perspective

**Chakkrit Luangcharoenrat** [1],*, **Singh Intrachooto** [1], **Vachara Peansupap** [2] **and Wandee Sutthinarakorn** [3]

1   Faculty of Architecture, Kasetsart University, Bangkok 10900, Thailand
2   Faculty of Engineering, Chulalongkorn University, Bangkok 10330, Thailand
3   Faculty of Education, Kasetsart University, Bangkok 10900, Thailand
*   Correspondence: cluang@gmail.com; Tel.: +66-81-845-9456

**Abstract:** Rapid growth in construction activities as a result of a growing population and urbanization in many parts of the world generates a large amount of waste from construction. To reduce and manage these wastes, a comprehensive understanding of the construction waste generation factors is needed. The purpose of this study is to identify the contributing factors of construction waste in Thailand's construction industry. The causes of construction waste were identified through an extensive literature review. A total of 28 causes of construction waste were identified and grouped into the four categories: design and documentation, material and procurement, construction method and planning, and human resources. To determine the significant level of each factor, a structured questionnaire survey was carried out to gather information from contractors about causes of construction material waste. The results show that the categories contributing to construction waste ranks as design and documentation, human resources, construction methods and planning, and material and procurement, respectively. Meanwhile, factors from each category were also determined and ranked. Design change, inattentive working attitudes and behaviors, ineffective planning and scheduling, and material storage were among the highest impact factors on construction waste generation in each category. Identifying the significance levels of waste generation factors will help the industry's stakeholders build suitable strategies to manage construction waste more effectively.

**Keywords:** construction waste; waste management; construction waste factors; sustainability; relative important index

## 1. Introduction

In 2017, the world population was 7.6 billion, and by 2050, this will increase to 9.8 billion people [1]. The increase in the world population causes problems such as overdrawn use of water, urban sprawls, increasing use of chemical substances, loss of natural forests, and the increase of motor vehicles. There is a significant increase in the use of oil, gas, and coal that leads to a rapid increase in carbon dioxide and methane in the Earth's atmosphere [2].

Currently, 55% of the world's population lives in urban areas, and this will increase to 68% by 2050. Urbanization will be required to meet the demands on transport, housing and energy supply, infrastructure, and waste management. In 2050, there will be another 2.5 billion people living in urban areas, with 90% of this migration occurring in Asia and Africa [3]. The population in the major cities of Thailand increased from 31.39% in 2000 to 49.95% in 2018. The population is likely to continue increasing, with as high as 69.46% of the people living in urban areas [3]. The construction industry is a major mechanism for infrastructure development to support cities' expansion and has a contributing role in environmental degradation. The construction industry uses 35% of produced energy and

released 40% of carbon dioxide into the Earth's atmosphere. At the same time, the construction industry is the largest consumer of raw materials derived from natural resources [4]. In addition, the building construction process also produces material waste that negatively impacts the environment.

Researchers have been collecting the amount of waste from construction projects in order to gain insight into the status of the problems and find ways to manage them. The proportion of construction debris (by weight) that is landfilled in each country shows between 13% and 60% compared with the total amount of waste (Table 1).

**Table 1.** Percentage of waste from construction compared with total construction waste.

| Country | Percentage of Waste | References |
|---|---|---|
| England | 32% | Sharman [5] |
| Hong Kong | 28% | EPDHK [6] |
| Netherland | 28% | |
| Australia | 20–30% | |
| United States | 20–29% | Bossink and Brouwers [7] |
| Germany | 19% | |
| Finland | 13–15% | |
| Chili | 34% | Mager [8] |
| Brazil | 50% | Contreras et al. [9] |
| Denmark | 27% | Katz and Baum [10] |
| Israel | 60% | Allwood et al. [11] |
| Japan | 20% | Yonetani [12] |
| Canada | 27% | Yeheyis et al. [13] |

The reduction of waste from construction will have numerous benefits, including natural resource conservation and reducing the use of virgin materials to produce construction materials, cost reductions from reducing the amount of construction materials, and reducing expenses from waste disposal [14]. Also, reducing waste creates a competitive advantage for stakeholders in the construction industry especially subcontractors, main contractors, and real estate developers. Other benefits derived from reducing waste are reducing carbon dioxide emissions ($CO_2$), reducing health problems in workers and communities around construction sites, prolonging landfill life spans, and reducing the cost of the project [15].

The purpose of the paper is to determine the significant level of factors in construction waste generation in Thailand. This paper also reviews previous studies on construction waste to provide a comprehensive overview of the current construction waste situations.

## 2. Literature Review

### 2.1. Reduce, Reuse, and Recycle

United Kingdom, North America, Europe, and various parts of Asia accepted the 3R Principle, which is to reduce, reuse, and re-process waste [11]. Economic and environmental benefits were the reasons to support the 3R Principle in building construction [16]. The environmental benefit includes prolonging landfill life span, reducing the use of raw material, reducing the environmental impact from raw material extraction and new material production processes, extending the lifespan of natural resources, and reducing environmental pollution harmful to human health and well-being. The economic advantages include reducing project costs, increasing business opportunities, reducing litigation risk relating to waste, and demonstrating the commitment to reduce the environmental impact of construction [17,18].

### 2.2. Construction Waste

Construction waste is different from municipal waste and typically comes from renovation, construction, modification and demolition of roads, buildings, and other built facilities [19].

Construction and demolition (C & D) debris is waste produced in the process of construction, renovation, or demolition of structures. Components of C & D debris include concrete, asphalt, wood, metals, gypsum wallboard, and roofing [20]. Waste generated from construction and demolition for England is the only waste generated from the construction site [21]. In Australia, construction wastes are caused by building demolition such as building construction, road construction, and railway construction and maintenance, including digging [22]. In Hong Kong, waste from construction was everything that occurred as a result of construction activities and that was left on construction sites, whether it was used or stored [23]. Waste from construction in Hong Kong was divided into two groups: (1) inert construction waste, most of which consists of construction materials, stone fragments, soil, asphalt, and concrete, which can be used to adjust the construction area; and (2) non-inert construction waste, which accounted for 20% of all construction waste, and consists of bamboo, wood, plants, packaging, and other organic materials. Some parts could be recycled, and some were disposed of and sent to a landfill. Waste from construction materials in this paper means various construction materials that cannot be reused, leftover construction material, and material damaged during construction or handling.

## 2.3. Types of Waste from Construction Materials

Waste from construction and demolition is considered high volume when compared with other types of waste, and causes environmental and social problems. The composition of construction waste is often unique because it depends on the construction techniques, building types, countries, and other factors. Construction techniques and varying building technologies cause difficulties in determining the type of waste from construction and demolition. However, there is an ongoing effort to determine the type or classification of C & D waste. Waste from construction will consist of inert waste and non-inert waste arising from construction [24]. Katz and Baum [10] indicated that the construction work could be divided into three periods according to the nature of waste from the construction such as the structural frame, the early finishing works, and the late finishing works (Figure 1). The study found the relationship between the amount of construction waste and the construction period. The amount of waste generated during structural frame construction was less than other periods.

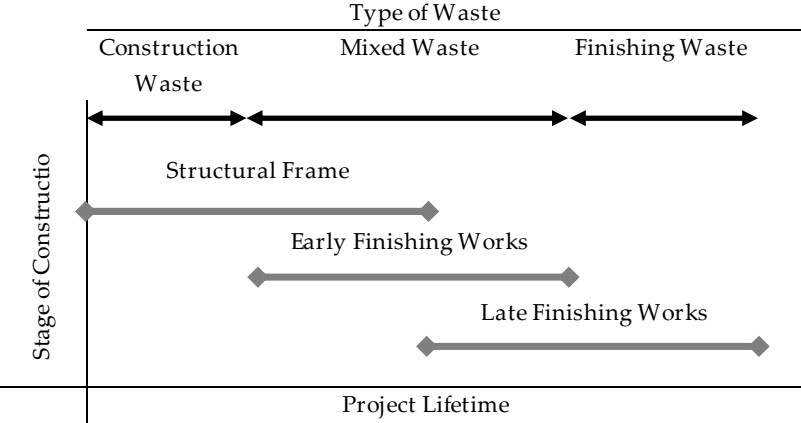

**Figure 1.** Construction process and waste type. Source: modified from Katz and Baum [10].

## 2.4. Construction Waste Categories

The analysis of construction waste compositions is important in helping to manage these wastes effectively. The European Union [25] divides the construction waste into eight categories: (1) brick, concrete, ceramic, and tile; (2) glass, wood, and plastic; (3) coal and asphalt, (4) metals; (5) soils, including soil excavated from contaminated site, rocks, and soils obtained from dredging; (6) insulation materials and materials containing asbestos, (7) construction materials containing gypsum, and (8) waste from other construction. England [26] has ten categories of construction waste: (1) insulation

and asbestos materials; (2) concrete, brick, tile, and ceramic; (3) wood, glass, and plastic; (4) asphalt, oil, coal, and bitumen; (5) metals; (6) soil, contaminated soil, stone, and soil from dredging; (7) gypsum; (8) cement, (9) paint and coating materials; and (10) glues and fillings. USEPA [27] divides waste from construction into 15 groups, including (1) asphalt-related materials, (2) soil related materials, (3) materials related to electrical works, (4) materials related to insulation, (5) materials related to bricks and concrete, (6) material related to steel, (7) materials related to paint work, (8) paper-related materials, (9) materials related to petroleum products, (10) materials related to roofing works, (11) materials related to vinyl, (12) gypsum related materials, (13) wood related materials, (14) materials related to wood containing contaminants, and (15) miscellaneous groups (Table 2).

**Table 2.** Summary of construction waste category.

| | EU [25] | England [26] | United States [27] |
|---|---|---|---|
| Asphalt | √ | √ | √ |
| Soil | √ | √ | √ |
| Electrical work | | | √ |
| Insulation | √ | √ | √ |
| Brick and concrete | √ | √ | √ |
| Steel | √ | √ | √ |
| Cement | | √ | |
| Paint | | √ | √ |
| Paper | | | √ |
| Petroleum | | √ | √ |
| Roof | | | √ |
| Vinyl | | | √ |
| Gypsum | √ | | √ |
| Wood | √ | √ | √ |
| Contaminated Wood | | | √ |
| Glues and fillings | | √ | |
| Miscellaneous | | | √ |

Researchers in both developing and developed countries studied construction waste compositions to properly manage and mitigate these wastes. Table 3 presents past research on construction waste compositions.

**Table 3.** Construction waste compositions research between 1994–2011.

| Research | Year | Country | Type of Building |
|---|---|---|---|
| Gavilan and Bernold [28] | 1994 | USA | Residential |
| Bossink and Brouwers [7] | 1996 | Netherlands | Residential |
| Franklin [20] | 1998 | USA | Residential |
| Poon et al. [29] | 2001 | Hong Kong | Not specific |
| Begum et al. [30] | 2006 | Malaysia | Residential |
| Uyasatean and Utwarujikulchai [31] | 2007 | Thai | Residential |
| Bergsdal et al. [32] | 2008 | Norway | Residential |
| Lau et al. [33] | 2008 | Malaysia | Residential |
| Kofoworola and Gheewala [34] | 2009 | Thai | Not specific |
| Guzmán et al. [35] | 2009 | Spain | Residential |
| Llatas [36] | 2011 | Spain | Residential |

Gavilan and Bernold [28] discussed development steps in developing a complete construction waste management system, classification, and measuring the amount of waste from construction. A source of waste framework was used to evaluate several residential buildings, and the three most common types of waste were brick and block, dimensional lumber, and sheetrock. Bossink and Brouwers [7] analyzed and quantified waste during building construction projects. The average

amount of waste from construction materials was between 1% and 10% by weight. Franklin's report [20] presented information about the amount and composition of C & D waste and management guidelines. The report was based on waste from renovation, demolition, and construction of buildings. Poon et al. [29] addressed construction waste problems in Hong Kong. Hong Kong produced C & D waste of about 32,710 tons/day in 1998. To deal with a large waste quantity, Hong Kong sent the inert (e.g., concrete, bricks, and sand) to public filling areas and the non-inert portion (e.g., wood, plastics, and paper) to municipal solid waste landfills. Standard compositions of C & D waste could improve waste management strategies. Begum et al. [30] presented the case studies on construction waste generation factors, compositions of C & D waste, and materials' reuse and recycling on the building construction sites. Uyasatean and Utwarujikulchai [31] assessed the amount and composition of C & D waste in Bangkok to improve C & D waste assessment guidelines. The study identified that the main compositions of waste from construction were concrete, brick, and steel. Bergsdal et al. [32] presented steps to estimate the amount of waste that would occur in the future through the stock and flow mathematical model. The case study indicated that in the future the C & D waste amount will be comprised of concrete, large brick, and wood. This forecast would be very useful in preparing measures to handle future waste. Lau et al. [33] evaluated the quantity and classification of C & D waste of residential buildings in Malaysia to build awareness and increase the construction waste segregation and the opportunity for recycling and reuse. Kofoworola and Gheewala [34] studied construction waste management in Thailand. The Thai construction industry produced an average of 1.1 million tons of waste from construction per year, representing 7.7% of the total waste amount that was sent to landfills. Guzmán et al. [35] described the waste management model called the Alcores, which was developed based on the study of 100 residential projects. This model could be used to estimate the expected waste amount that will occur during building construction. Llatas [36] presented a model that will allow technicians to estimate the waste amount from C & D in design stage in order to find methods to prevent and reduce waste. Types and C & D waste amount are evaluated under the EU guidelines by creating an appropriate composition for each project. This model helps to detect waste sources and can be used as alternative steps to reduce hazardous waste and reduce waste from construction and demolition. Table 4 presents parentage of construction waste material gathering from the literature. Concrete, brick, and wood were the top three material wastes that were commonly found and were greater in amount when compared with other materials on the construction site.

### 2.5. Construction Material Waste Generation Factors

Waste generation factors from construction activities vary depending on the size of the project, related activities, and the project location. The construction waste may arise from the beginning of the construction process, such as site clearing, through project handover. Gavilan & Bernold [28] found that design, procurement, material handling, operation, and leftover scraps on site were major causes of waste. They further proposed the site construction material flow model along with the guidelines for dealing with those wastes (Figure 2).

**Table 4.** Composition of construction waste in various countries.

| Type of Construction Waste | Gavilan and Bernold [28] | Bossink and Brouwers [7] | Franklin [20] | Poon et al. [29] | Begum et al. [30] | Uyasatean and Utwarujikulchai [31] | Bergsdal et al. [32] | Lau et al. [33] | Kofoworola and Gheewala [34] | Guzmán et al. [35] | Llatas [36] |
|---|---|---|---|---|---|---|---|---|---|---|---|
| Concrete | | 13% | 8% | 18% | 57% | 77% | | 24% | | 8% | 12% |
| Brick | 12% | 3% | 2.5% | 6% | 12% | 14% | 46% | 14% | 46% | 8% | |
| Wood | 35% | | | 11% | 7% | | 14% | 56% | 14% | | 4% |
| Dirt/Earth | | | 0.5% | 31% | | | | | | 65% | |
| Tile/Ceramic | | | 1% | | 1% | 3% | | | | 4% | 54% |
| Drywall | 15% | | 14% | | | | 6% | | 6% | 4% | |
| Stone | | 29% | | | 1% | | | | | | 5% |
| Rebar | | | | | 22% | 5% | | | | | |
| Other Paper | 4% | 7% | | | | | 5% | | 5% | | |
| Piles | | 17% | | | | | | | | 3% | |
| Metal | 4% | | 1% | 4% | | | 1% | 2% | 1% | | 3% |
| Rubber | | | | 14% | | | | | | | |
| Mortar | | 8% | | | | | | | | | |
| Untreated Wood | | | 16% | | | | | | | | |
| Misc. Waste | 30% | 23% | 57% | 16% | | 1% | 28% | 4% | 28% | 8% | 22% |
| Total | 100% | 100% | 100% | 100% | 100% | 100% | 100% | 100% | 100% | 100% | 100% |

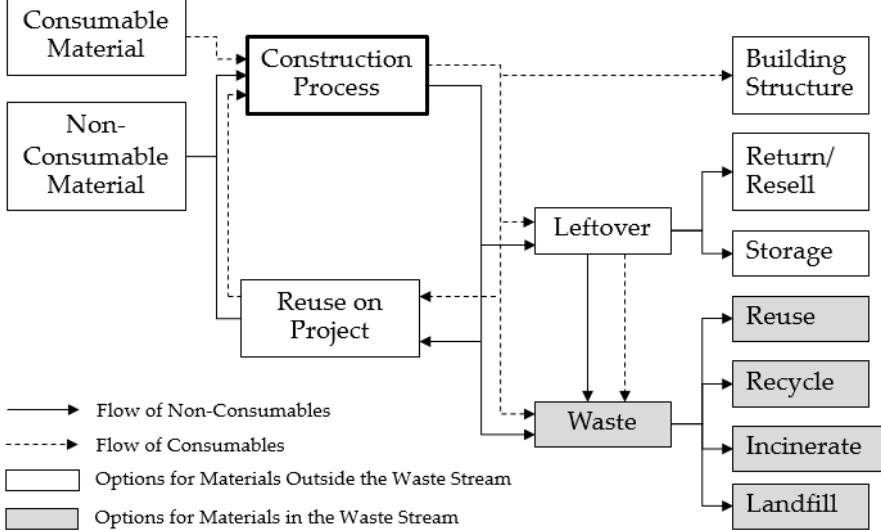

**Figure 2.** Generic flow pattern of construction material on site. Source: adapted from Gavilan & Bernold [28].

The literature review showed a large number of past studies that were conducted to identify the construction waste generation factors. Faniran and Caban [37] studied the construction waste reduction strategy using a survey method of construction companies. The research found that many companies did not have a specific policy to reduce waste, while companies with clear policies had tried to reduce the waste at the source, such as avoiding the generation of waste from the construction process. The results of the survey identified that the five major causes of construction waste include changes in design, leftover material, packaging waste, errors in design or detailing, and poor weather condition. Alwi et al. [38] studied waste generation rates in construction companies in Indonesia and found six factors to be the key variables of construction waste generation, including repair works, waiting time for materials, schedule delay, non-skilled workers, raw materials waste on-site, and lack of supervision. The Singaporean building construction industry experienced that design, operation, and material handling were critical site waste sources [39]. The study of the construction waste causes during design and construction in the U.K. [40] found that last minute changes had the highest rating by both architects and contractors. Wan et al. [41] studied the waste sources at each stage of electrical and mechanical engineering works in Hong Kong and found that poor coordination and design changes and errors were major contributors to variations or change orders and rework. Contractors' perceptions on the sources of waste in the United Arab Emirates were lack of awareness, off-cuts resulting from poor design, and reworks and variations [42]. Nagapan et al. [43] conducted research in Malaysia and found that the five key factors of construction waste were inadequate site management and supervision, lack of work experiences, poor planning and scheduling, design mistakes and errors, and mistakes during construction process. Muhwezi et al. [44] reported the waste causes in construction in Uganda included design changes during construction, lack of skilled workers or sub-contractors, non-compliant products, inappropriate material storages, changing orders/instructions. The study by Mbote et al. [45], which conducted research in Kenya, showed that complex or poor designs, inadequate security, poor work conditions, and topography were the major causes of construction waste. John and Itodo [46] showed that inefficient job control, re-work, and bad material management were the important factors contributing to high material waste. Adewuyi and Otali [47] assessed the factors causing waste from construction in Nigeria. The results indicated that the three most important factors contributing to material waste in construction were rework, design changes, and waste from unusual shapes and forms. The five most important waste factors of the Vietnam construction industry, in high-rise building projects, were supervising and inspecting time, waiting for others treads, accidents, workers transporting time, equipment and materials, and workers' rest time during construction [48].

A study by Fadiya et al. [49] found nine sources of construction waste in the U.K. construction industry. These were residual, design, handling, data error, operations, others, misplacement, weather, and vandalism. Bekr [50] indicated that the most important causes of materials wastage on construction sites were design changes, rework, poor documents, improper and inadequate of materials storage, poor waste management strategy, lack of skilled workers, bad site conditions, damaged material during transportation, mistakes in quantity estimation and over allowance, and theft and vandalism. Domingo [51] conducted research on the generation of construction waste factors in construction. The findings revealed that incorrect information on drawings, incomplete briefing, complicated designs, non-standard designs, and poor coordination in the building construction lifecycle were the construction waste causes.

## 2.6. Categorization of Construction Material Waste Generation Factors

Scholars around the world have studied waste in building construction for decades. Significant factors that contribute to construction waste generation were found and grouped. A review of past research and literature indicated categorization of the various factors in groups of up to nine categories including design, procurement, construction methods, equipment, material, labor and human behavior, owner, project, and weather. This study re-clustered these factors into four categories: design and documentation (DEDO), material and procurement (MAPR), construction method and planning (COPL), and human resources (HUMA).

## 2.7. Design and Documents(DEDO)

Design is the initial step in the development of construction projects before entering the construction process. Some causes of construction waste are the lack of attention of the designers in the construction process and constructability of design intention. Specifying too many materials and sizes in the construction project may lead to ordering large amounts of material because of minimum order or production requirements from the suppliers. This material cannot be used in actual construction and may remain on site and end up as waste. Designs not taking standard sizes into consideration may generate waste due to cutting to fit the shape or size of an installed area. The lack of knowledge of the standard size of the actual building materials on the market is also the cause of construction waste [52] and the generation of waste in the design phase is caused by providing flaw details and changes [28]. The design and documents play an important role throughout the construction process.

## 2.8. Material and Procurement (MAPR)

Gavilan and Bernold [28] stated that the construction waste generation factors from the procurement process could be (1) ordering more materials than the actual amount needed, (2) ordering materials less than the actual amount used as a result of miss estimation, and (3) ordering the wrong materials as a result of communication or information errors. The study by Karim and Marosszeky [52] indicated that inappropriate material handling and storage caused construction waste. Inadequate handling and improper transportation were the major sources of material waste [53]. Transportation of material from manufacturing to construction site or within site without procedure could damage the material and later turn it to waste. Ajayi et al. [54] suggested that four features characterized a waste efficient procurement process and logistics: suppliers' participation in low waste measures, material waste purchase management, effective materials management, and waste efficient bills of quantity.

## 2.9. Construction Methods and Planning (COPL)

Complexity is a key characteristic of construction projects because of building structure and function, construction method, the project schedule, size/scale of project, site conditions, and the neighboring environment [55]. The complexity degree determines the overall approach to a project, specifically the required resources and planning, as well as the tools and techniques. Each construction

project is unique and complex and requires a tremendous amount of work to be done. These works have directly affected the waste generated amount by construction activities. Karim and Marosszeky [52] indicated that waste during the construction phase was largely caused by pre-construction preparations, but there were still many other factors that caused waste in the construction process. Waste during the construction process could be generated as the result of bad coordination and control [38,43,44,48], wrong choice of construction methods [7,41], and reworks [46,50].

### 2.10. Human Resources (HUMA)

Construction is considered a labor-intensive industry. Effective management and reduction of waste from construction depends on cooperation, attitudes, and behaviors of people involved in the construction process. Skoyles et al. [56] indicated that the level of waste from construction depended on labor and personnel factors rather than other factors. Workers who are not taught and trained, lack skills in assigned tasks, and have a bad attitude will affect the quality of work and then cause rework and repairs. Ali et al. [57] indicated that the factors affecting time extension of building construction included the lack of experienced consultants, engineers, and staff. Designers who lack experience may cause design changes during construction as a result of misinterpretation of the needs of project owners and stakeholders and choosing the wrong materials or construction methods. Lack of knowledge of the designer regarding the materials and construction equipment was the most crucial reason that the design and construction did not synchronize and caused work problems [58]. Barrett and Stanley [59] stated that designers should have the skill to capture and understand all project requirements and to transfer this information to the construction site through construction drawings.

## 3. Research Methodologies

Research is being developed to examine contractors' points of view and the levels of importance among construction waste generation factors.

### 3.1. Instrument

A Google Forms-based questionnaire is used to collect information from contractors, architects, and construction managers in Thailand. The completed questionnaires could be submitted via email, and then were automatically converted and stored in MS Excel. The process of developing the questionnaire was as follows:

1.  A comprehensive list of 28 construction waste generation factors that were identified through the literature. The literature review identified 166 factors that affect construction waste. These factors were cross referenced and reduced to 28 based on the following two steps that were suggested by Wambeke et al. [60]: (1) removing factors that were only suitable for specific projects and tasks, and (2) combining similar factors under the same category. These causes were grouped into four categories according to their sources. Factors that cause construction waste in this paper are shown in Table 5 below.
2.  The content validity and appropriateness of the questions in the questionnaire were reviewed by academics and construction practitioners to verify comprehensibility and relevancy of the contents.
3.  A pilot survey was conducted to test the reliability of the content and the design of the survey. The questionnaires were sent out to a sample group of thirty construction personnel. The total internal consistency reliability coefficients were 0.985.
4.  The final questionnaire is divided into two parts. The first part covers the general background of the respondents. The second part covers respondents' reflection on the levels of influence/importance/significance of the construction waste generation factors within each of their building construction projects. Respondents are asked to rate their levels of agreement to a statement on five levels: (1) strongly disagree; (2) disagree; (3) neither agree nor disagree; (4) agree; (5) strongly agree.

**Table 5.** Factor categories, factor labels, and factor name.

| Factor Category | Factor Label | Factor Name | References |
|---|---|---|---|
| Design and Documentation (DEDO) | D1 | Change to design | [7,28,37,38,40,41,43,47,49–51] |
| | D2 | Document problems | [7,38–40,44,47–50] |
| | D3 | Design errors | [28,37,38,40,42,43,46] |
| | D4 | Construction drawing errors | [28,38–40,44,47,51] |
| | D5 | Complicated design | [39,44,47,50,51] |
| Material and Procurement (MAPR) | M1 | Improper material storage | [7,28,37–42,46–51] |
| | M2 | Material quality problems | [7,38,39,42,44,47,48,50] |
| | M3 | Material ordering problems | [7,37,39,41,42,47,49,50] |
| | M4 | Improper material handling | [28,37,38,42,46–48,50] |
| | M5 | Material transporting problems | [28,39,42,47,49–51] |
| | M6 | Packaging problems | [28,39,42,46,50] |
| | M7 | Defective materials | [41,47] |
| | M8 | Damaged materials | [47] |
| Construction Methods and Planning (COPL) | C1 | Coordination problems | [28,38–41,43,44,47,48,50,51] |
| | C2 | Control and supervision | [37,38,43,44,46–49] |
| | C3 | Construction methods | [28,38,40,41,47–49] |
| | C4 | Poor waste management | [28,44,46,47,49,50] |
| | C5 | Tools and equipment misuse/malfunction | [7,28,39,47,49] |
| | C6 | Misuse of material | [28,38,39,47,48] |
| | C7 | Rework | [41,43,46,47,50] |
| | C8 | Wrong teams /subcontractors selection | [38,41–43,47] |
| | C9 | By-process waste | [7,28,37,40] |
| | C10 | Construction errors | [41,43,46] |
| | C11 | Ineffective planning and scheduling | [38,41,43,44,47,48,51] |
| Human Resources (HUMA) | H1 | Incompetent workers | [7,37–41,44,46–50] |
| | H2 | Designers' inexperience | [7,39,40,43,44,46,50] |
| | H3 | Inattentive working attitudes and behaviours | [39,41,44,50] |
| | H4 | Lack of suppliers' involvement | [37,47,50] |

### 3.2. Sampling

The distribution of the questionnaire was directed toward general contractors. The survey was sent in two different ways: via direct e-mail through construction companies listed on the Ministry of Commerce in Thailand website and via Line and Facebook of various contractor groups. It was difficult to determine the sample size because of distribution methods. The survey was open for ten months between March and December 2017. There were all together 178 survey respondents.

### 3.3. Analysis Techniques

Relative importance index (RII) was used for generating an index because this method is able to rearrange the factors being studied [61]. Othman et al. [62] used RII to determine the relative importance of factors that drive changes to the construction project brief. Gunduz et al. [63] and Aziz [64] used RII to rank delay factors in construction projects. The same method was adopted in this study.

The Statistic Package for Social Science (SPSS) was used where the scores assigned to each factor by the respondents were entered. Then, the response questionnaires were subjected to statistical analysis. The contribution of each of the factors to construction waste generation was evaluated and the ranking of the attributes in terms of their important as perceived by the respondents was done using RII, which was computed using following Equation (1).

$$\text{RII} = \Sigma W / (A \times N), \ (0 < \text{RII} < 1), \tag{1}$$

where W represents the weight that gives to each factor by the respondents and ranges from 1 to 5; (1) is strongly disagree and (5) is strongly agree; A represents the highest weight (i.e., 5 in this case); and N represents the total number of respondents.

## 4. Results

The following subsections describe the survey results: (a) an overview of the respondents and (b) relative important index (RII) of construction waste generation factors.

### 4.1. Respondents

A total of 178 responses were used for the analysis. Table 6 presents respondents' and construction project characteristics. The demographic data highlighted that wide-ranging positions in construction companies were represented. The majority of building types surveyed are residential projects (private residential building, condominium, and apartment). They are predominantly buildings that have construction areas greater than 4000 m$^2$. Fifty percent of the buildings under construction are lower than 15 m in height. These characteristics provide insight into construction waste generation factors that are influenced by building type, size, and height, as well as whether it differs.

**Table 6.** Characteristics of survey respondents (n = 178).

| Characteristic | | Frequency |
|---|---|---|
| **Gender** | Male | 162 (91%) |
| | Female | 9 (9%) |
| **Age group** | Under 30 | 26 (14.61%) |
| | 30 to 40 | 29 (16.29%) |
| | 41 to 50 | 77 (43.26%) |
| | 50 and over | 46 (25.84%) |
| **Position** | Managing director | 5 (2.81%) |
| | Project manager | 56 (31.46%) |
| | Project engineering | 38 (21.35%) |
| | Project director | 18 (10.11%) |
| | Assistant project manager | 2 (1.12%) |
| | Site engineering | 16 (8.99%) |
| | Project architecture | 14 (7.87%) |
| | Site architecture | 5 (2.81%) |
| | Foreman | 24 (13.48%) |
| **Building size (m$^2$)** | <500 | 36 (20.22%) |
| | 500–2000 | 23 (12.92%) |
| | 2001–4000 | 22 (12.36%) |
| | 4001–10,000 | 32 (17.98%) |
| | 10,001–30,000 | 46 (25.84%) |
| **Experience in construction** | <5 years | 22 (12.36%) |
| | 5–10 years | 42 (23.60%) |
| | 11–15 years | 66 (37.08%) |
| | 16–20 years | 31 (17.42%) |
| | 20 years< | 17 (9.55%) |
| **Education** | Vocational | 38 (21.35%) |
| | Bachelor | 83 (46.63%) |
| | Master | 56 (31.46%) |
| | Ph.D. | 1 (0.56%) |
| **Building type** | Private Residential | 40 (22.47%) |
| | Condominium & apartment | 50 (28.09%) |
| | Public building | 53 (29.78%) |
| | Industrial | 15 (8.43%) |
| | Government | 20 (11.24%) |
| **Building height (Meter)** | <15 | 89 (50.00%) |
| | 15–23 | 41 (23.03%) |
| | 23< | 48 (26.97%) |

### 4.2. Relative Important Index (RII)

Table 7 represents the respondents the total number for each selection per evaluated factor and for analyzing data. The relative important index (RII) technique was used per factor. The index was computed using Equation (1). Table 8 show the mean RII and the ranking of all categories.

**Table 7.** Relative important index (RII) and ranking of construction waste generation factors (n = 178).

| Factor Label | Factor | Number of Respondents Scoring | | | | | RII | Rank |
|---|---|---|---|---|---|---|---|---|
| | | 1 | 2 | 3 | 4 | 5 | | |
| D1 | Change to design | 2.00 | 22.00 | 50.00 | 70.00 | 34.00 | 0.726 | 1 |
| H3 | Inattentive working attitudes and behaviours | 9.00 | 35.00 | 42.00 | 61.00 | 31.00 | 0.679 | 2 |
| M1 | Improper material storage | 11.00 | 28.00 | 51.00 | 65.00 | 23.00 | 0.669 | 3 |
| H2 | Designers' inexperience | 7.00 | 42.00 | 45.00 | 54.00 | 30.00 | 0.665 | 4 |
| H1 | Incompetent workers | 6.00 | 43.00 | 47.00 | 52.00 | 30.00 | 0.664 | 5 |
| D5 | Complicated design | 9.00 | 36.00 | 53.00 | 51.00 | 29.00 | 0.662 | 6 |
| D3 | Design errors | 6.00 | 46.00 | 48.00 | 51.00 | 27.00 | 0.653 | 7 |
| C11 | Ineffective planning and scheduling | 9.00 | 39.00 | 63.00 | 38.00 | 29.00 | 0.644 | 8 |
| C2 | Control and supervision | 10.00 | 42.00 | 62.00 | 36.00 | 28.00 | 0.634 | 9 |
| C4 | Poor waste management | 20.00 | 31.00 | 46.00 | 61.00 | 20.00 | 0.634 | 9 |
| C8 | Wrong teams/subcontractors selection | 12.00 | 46.00 | 52.00 | 38.00 | 30.00 | 0.631 | 10 |
| M3 | Material ordering problems | 11.00 | 42.00 | 55.00 | 53.00 | 17.00 | 0.626 | 11 |
| D4 | Construction drawing errors | 13.00 | 51.00 | 42.00 | 46.00 | 26.00 | 0.624 | 12 |
| M4 | Improper material handling | 13.00 | 40.00 | 56.00 | 54.00 | 15.00 | 0.620 | 13 |
| D2 | Documents problems | 8.00 | 52.00 | 54.00 | 44.00 | 20.00 | 0.618 | 14 |
| C3 | Construction methods | 13.00 | 50.00 | 46.00 | 49.00 | 20.00 | 0.615 | 15 |
| M5 | Material transporting problems | 13.00 | 42.00 | 61.00 | 49.00 | 13.00 | 0.608 | 16 |
| C7 | Reworks | 14.00 | 45.00 | 59.00 | 40.00 | 20.00 | 0.608 | 16 |
| C9 | By Process waste | 20.00 | 40.00 | 55.00 | 50.00 | 13.00 | 0.596 | 17 |
| C1 | Coordination problems | 19.00 | 52.00 | 50.00 | 37.00 | 20.00 | 0.585 | 18 |
| C5 | Tools and equipment misuse/malfunction | 16.00 | 48.00 | 63.00 | 35.00 | 16.00 | 0.585 | 18 |
| M7 | Defective materials | 18.00 | 47.00 | 56.00 | 49.00 | 8.00 | 0.580 | 19 |
| C10 | Construction errors | 21.00 | 53.00 | 49.00 | 35.00 | 20.00 | 0.578 | 20 |
| M6 | Packaging problems | 20.00 | 53.00 | 50.00 | 46.00 | 9.00 | 0.567 | 21 |
| H4 | Lack of suppliers involvement | 22.00 | 55.00 | 50.00 | 38.00 | 13.00 | 0.561 | 22 |
| M8 | Damaged materials | 24.00 | 57.00 | 48.00 | 36.00 | 13.00 | 0.552 | 23 |
| M2 | Material quality problems | 33.00 | 53.00 | 42.00 | 34.00 | 16.00 | 0.540 | 24 |
| C6 | Misuse of material | 38.00 | 52.00 | 42.00 | 28.00 | 18.00 | 0.528 | 25 |

**Table 8.** Mean RII and ranking of categories of construction waste generation factors (n = 178).

| Factor Category | RII | Rank |
|---|---|---|
| Design and Documentation (DEDO) | 0.656 | 1 |
| Human Resources (HUMA) | 0.642 | 2 |
| Construction Methods and Planning (COPL) | 0.603 | 3 |
| Material and Procurement (MAPR) | 0.595 | 4 |

### 4.2.1. Design and Documentations (DEDO)

The RII and ranks of the five (5) factors that are grouped under the "DEDO" are shown in Table 7. The respondents ranked the "change to design" factor as the most contributing factor for construction waste generation, with RII equal to 0.726. Change to design ranked first in its effect, among all explored factors, which shows the essential impact on the construction waste generation.

### 4.2.2. Human Resource (HUMA)

The RII and ranks of the four factors that are grouped under the "HUMA" are shown in Table 7. The respondents ranked the "inattentive working attitude and behaviors" factor as the most contributing factor for construction waste generation, with RII equal to 0.679. Inattentive working attitude and behaviors ranked second in its effect, among all explored factors, which shows the essential impact on the construction waste generation. Three out of four factors in this category are also ranked in the top five overall factors that have a significant impact on construction waste generation.

### 4.2.3. Construction Methods and Planning (COPL)

The RII and ranks of the eleven factors that are grouped under the "COPL" are shown in Table 7. The respondents ranked the "ineffective planning and scheduling" factor as the most contributing factor for construction waste generation, with RII equal to 0.644. Ineffective planning and scheduling ranked eighth in its effect, among all explored factors, which shows its lesser impact on the construction waste generation.

### 4.2.4. Material and Procurement (MAPR)

The RII and ranks of among the eight factors that are grouped under the "MAPR" are shown in Table 7. The respondents ranked the "improper material storage" factor as the most contributing factor for construction waste generation, with RII equal to 0.669. Improper material storage ranked third in its effect, among all explored factors, which shows its essential impact on the construction waste generation.

## 5. Discussion

A number of factors rated by the survey respondents with respect to their effect on construction waste generation have been identified and ranked according to their relative importance index (RII) in Table 7. The five factors with the highest RII were change to design, inattentive working attitudes and behaviors, improper material storage, designers' inexperience, and incompetent workers. This result implies that design change (which is related to owners' and designers' direct changes, and changes due to construction site conditions or mistakes) during construction is likely to be an important factor contributing to construction waste generation. Osmani et al. [40] estimated that about 33% of construction waste could arise from design decisions. Design changes during construction are major origins of construction waste production [7,37,65] and rework [66]. Designers who lack experience and knowledge about construction methods and techniques during the design process can also result in waste being produced [67]. Inattentive working attitudes and behaviors, designers' inexperience, and incompetent workers are human-related factors. Construction is a labor-intensive industry. Workers' behaviors are likely to significantly influence waste levels. Successful construction waste management and minimization depend on the individual's willingness involved in the construction to change their behavior and attitudes [30,52,68]. Designers', supervisors', and workers' competences are likely to be important in successful construction waste management, and the experience of these people is important in terms of controlling or reducing other waste generation factors. Improper material storage is the third highest RII among construction waste generation factors. This problem always includes unsuitable storing methods and inappropriate protection [65]. Proper storage of material is highly necessary to mitigate construction waste generation.

Table 8 represents mean RII and ranking of categories of construction waste generation factors. Design and documentation (DECO) are the leading category factors of construction waste generation, followed by the human resources (HUMA) factors category. Construction methods and planning (COPL) and material and procurement (MAPR) are surprisingly the categories of low impact. Adewuyi and Otali [47] and Fadiys et al. [49] also confirmed that design and documentation was the leading source of construction waste. On the basis of the research of Ekanayake and Ofori [39], design, operational, procurement, and material handling contribute to increase waste on the construction site.

## 6. Conclusions

To minimize construction waste in building projects, major factors contributing to increase waste must be recognized. This research identified and determined the ranking among twenty-eight (28) factors of building construction waste generation. The explored factors were classified under the following four primary classifications: (1) design and documentation (DEDO); (2) human resources (HUMA); (3) construction methods and planning (COPL); and (4) material and procurement (MAPR).

The relative importance indices of construction waste generation factors were quantified and ranked and grouped according to their importance levels.

The findings of this study show that design and documentation-related factors are the major contributor to construction waste generation. Change to design, complicated design, and design errors are the top three factors in this category. Designers play crucial roles in preventing waste from the beginning of design stage through construction completion. Clearly defined drawings and documents reduce the discrepancies on construction documents, resulting in fewer changes and rework during construction. Human-related factors are ranked as the second major contributor to construction waste generation. Inattentive working attitudes and behaviors and designers' inexperience are the top two factors in this group and are also ranked in the top five among the overall factors. Attitude, behavior, and expertise of construction process participants toward construction waste management are important to minimize waste during construction. Construction methods and planning and material and procurement related factors are ranked third and last, respectively. For effective waste minimization of building construction in Thailand, this study recommends that all stakeholders, not only contractors and sub-contractors, in the construction industry should address these factors at every level of their construction processes and devise waste management plans. The focus should be (1) design and document management to make sure that provided information is clear and comprehensive enough for the construction stage, and (2) human management by having well-trained staff and workers in the field.

## 7. Recommendations

On the basis of the above-mentioned findings and interviews with contractors who had experience in supervising building construction, the following points are recommendations to minimize and manage waste in building construction projects. (1) The contractors should pay attention to construction documents and drawings to see any discrepancy and seek advice or answers from the owner or designers prior to construction; (2) The contractors must try to understand the owner's and designer's intention for the project to mitigate rework; (3) The contractors should have sufficient knowledge and expertise. They should gain experience before the construction stage so that they can seek out needed resources; (4) The amount and quality of materials on site should be checked and stored in the proper locations; (5) Site supervision and management should be done regularly, the lack of supervision may result in mistakes, reworks, and poor workmanship; (6) Coordination among parties is a key to reducing waste. Effective coordination can ease most of construction waste generation factors. Appropriated coordination among various parties should be formed for all phase of construction lifecycle. Finally, similar studies can be done with a focus on other types of construction projects to gain further insight into waste generation factors.

**Author Contributions:** Conceptualization, C.L., S.I., V.P. and W.S.; methodology, C.L., S.I., V.P. and W.S.; formal analysis, C.L.; data curation, C.L.; writing—original draft preparation, C.L.; writing—review and editing, C.L. and S.I.; visualization, C.L.

**Funding:** This research was funded by Kasetsart University under PhD. Scholarship Program.

**Conflicts of Interest:** The authors declare no conflict of interest.

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
