# Peer review of "Factors Influencing Construction Waste Generation in Building Construction: Thailand’s Perspective"

_sustainability, doi:10.3390/su11133638_

Reviewer 1 Report

Extensive editing of English language and style required and the authors must include the innovation of the study the author should identify the types of waste generated in order to be able to recycle.

Author Response

Dear Reviewer,

Thank you for your kind comments. 

Point 1: Extensive editing of English language and style required

Response 1: I had my American friend and my Prof., Ph.D. from MIT, review, and comment on my paper. Then, I revised paper based on their comments.

Point 2: The author must include the innovation of the study the author should identify the types of waste generated in order to be able to recycle

Response 2: Data on types of construction waste were not collected because it was not the research objective. I did have a section on types of construction waste that were based on literature review, so I touched on the type of construction waste that was commonly found in the construction site (line 171-172)

I will take your comments and do it on my next research.

Thank you for your time.

Chakkrit Luangchareonrat, DGNB

Reviewer 2 Report

Dear Authors,

Minimizing waste in the economy is an important issue around the world. The presented article is an example of this. I think that it is written correctly and is interesting. I have only one observation. The basic principle of the economic activity of man and its existence on earth is the generation of waste. Of course, minimizing the amount of waste is very important, but also the ways of their management are important. In other words, waste will not be avoided, so we should strive to develop methods for their use. What the authors wrote in the article, that in Thailand mainly used storage in relation to construction waste is very incorrect. He understands that the building waste management methods in Thailand were not the subject of the article, so please publish the article in the scientific journal.

Congratulations on the article 

Author Response

Dear Reviewer,

Thank you for your kind comments. 

Point 1: Minimizing waste in the economy is an important issue around the world. The presented article is an example of this. I think that it is written correctly and is interesting. I have only one observation. The basic principle of the economic activity of man and its existence on earth is the generation of waste. Of course, minimizing the amount of waste is very important, but also the ways of their management are important. In other words, waste will not be avoided, so we should strive to develop methods for their use. What the authors wrote in the article, that in Thailand mainly used storage in relation to construction waste is very incorrect. He understands that the building waste management methods in Thailand were not the subject of the article, so please publish the article in the scientific journal.

Response 1: Most of the information/data in this research were collected from the contractors that build the buildings in the Bangkok Metropolitan Region.

From my observation of these building construction project, there were not many spaces left for material storage because building owners would like to maximize site potentials such as have their building be built to meet the maximum floor area ratio and minimum open space due to land cost. As a result, limited storages space may effect improper stacking and store of material. So the material will be damaged before use and turn to waste later.

Based on this research, I will focus on each factor to find a suitable waste management strategies to minimize waste.

Thank you for your time.

Chakkrit Luangchareonrat, DGNB

Reviewer 3 Report

English language and style are fine/minor spell check required, but this is a well written paper. 

Author Response

Dear Reviewer,

Point 1: English language and style are fine/minor spell check required

Response 1: I will have my American friend and my Prof., MIT Ph.D., review and comment on my paper. Then, I revise paper based on their comments.

Thank you for your time.

Chakkrit Luangchareonrat, DGNB

This manuscript is a resubmission of an earlier submission. The following is a list of the peer review reports and author responses from that submission.

Round  1

Reviewer 1 Report

The article is poorly written.  It is difficult to understand clearly many sentences.  The text is redundant. 

From the way it is written  I cannot perform a proper review.

Reviewer 2 Report

Dear authors

Thanks for this interesting article and the good effort involved. I wold like to see major improvement made to the manuscript before it is ready for further assessment.

My points of concern are:

1) The quality of English really needs improvement. Mistake occurs as early as in the title of the paper. Elsewhere, please ensure that figure captions are correctly written; e.g. this caption "Factors from Human (HUMA) create waste from construction" has grammatical error.

2) The analysis of data lacks depth. More can certainly be done - e.g. evaluating whether there is any statistically significant difference between the scores of the different stakeholder groups interviewed. By giving only the descriptive statistics and summarized description of information gathered from the interviews are insufficient.

Reviewer 3 Report

Both the title and the abstracts do not reflect all the findings of the authors, it would be necessary to complement the summary and perhaps modify the title of the work.

The author does not emphasize the importance and influence of the human factor in the generation of waste in construction, perhaps this is the major contribution of this study.

In the section on conclusions it is necessary to highlight and link some concepts, findings and reasoning, which together could be part of the "culture of waste management in construction" such as:

- Waste related to people

- Measures to reduce waste

- Selection of waste at the construction site

- Nobody has the direct responsibility to classify waste

- Bad attitude and bad behavior of workers

- Motivation, cooperation, coordination

- Work plan, management system

- Staff training, lack of experience

- Morals, education, change

Reference within the text figure 1 and tables 2, 4, 5, 6, 7, 8, 9, 12 and 13

Increase size of figure 2 for better appreciation

Reviewer 4 Report

The article is interestin and  very current. However, the authors made it very casually. Below are the notes.

It is proposed to expand the article abstracts. Please describe the test results received. Note the lack of a capital letter at the beginning of the task.

No dot in line 21.

 There is no reference to Figure 1 in the article text.
How to understand the entry in line 57: "... reduce smuggling ..."?

Line 61: (CO2) should be the subscript.
Line 134: I suggest to exchange the word from "sent" to "forwarded".
Most tables have no reference in the article text!
Check punctuation errors throughout the article!

What does the abbreviation "THB" in table 13 mean? I understand that this is the currency of Thailand. However, I think that it is necessary to provide relations to the international currency, e.g. the US dollar.